# Diabetes-Induced Amplification of Nociceptive DRG Neuron Output by Upregulation of Somatic T-Type Ca^2+^ Channels

**DOI:** 10.3390/biom13091320

**Published:** 2023-08-28

**Authors:** Arsentii Ivasiuk, Maksym Matvieienko, Nikolai I. Kononenko, Dmytro E. Duzhyy, Sergiy M. Korogod, Nana Voitenko, Pavel Belan

**Affiliations:** 1Department of Molecular Biophysics, Bogomoletz Institute of Physiology of NAS of Ukraine, 01024 Kyiv, Ukraine; arsentii.ivasiuk@gmail.com (A.I.); ty4ka.m@gmail.com (M.M.); ni.kononenko@biph.kiev.ua (N.I.K.); isabroad@gmail.com (S.M.K.); 2Department of Sensory Signaling, Bogomoletz Institute of Physiology of NAS of Ukraine, 01024 Kyiv, Ukraine; dduzhyy@biph.kiev.ua; 3Department of Biomedicine and Neuroscience, Kyiv Academic University of NAS of Ukraine, 03142 Kyiv, Ukraine; 4Research Center, Dobrobut Academy Medical School, 03022 Kyiv, Ukraine

**Keywords:** calcium current, diabetic painful neuropathy, action potential, spinal cord, nociception, temperature dependence, excitability, simulations

## Abstract

The development of pain symptoms in peripheral diabetic neuropathy (PDN) is associated with the upregulation of T-type Ca^2+^ channels (T-channels) in the soma of nociceptive DRG neurons. Moreover, a block of these channels in DRG neurons effectively reversed mechanical and thermal hyperalgesia in animal diabetic models, indicating that T-channel functioning in these neurons is causally linked to PDN. However, no particular mechanisms relating the upregulation of T-channels in the soma of nociceptive DRG neurons to the pathological pain processing in PDN have been suggested. Here we have electrophysiologically identified voltage-gated currents expressed in nociceptive DRG neurons and developed a computation model of the neurons, including peripheral and central axons. Simulations showed substantially stronger sensitivity of neuronal excitability to diabetes-induced T-channel upregulation at the normal body temperature compared to the ambient one. We also found that upregulation of somatic T-channels, observed in these neurons under diabetic conditions, amplifies a single action potential invading the soma from the periphery into a burst of multiple action potentials further propagated to the end of the central axon. We have concluded that the somatic T-channel-dependent amplification of the peripheral nociceptive input to the spinal cord demonstrated in this work may underlie abnormal nociception at different stages of diabetes development.

## 1. Introduction

Peripheral diabetic neuropathy is one of the earliest, most frequent, and troublesome complications of diabetes, occurring in about 60–70% of diabetic patients [1]. Approximately 1/3 of these patients suffer from painful diabetic neuropathy (PDN), characterized by severe burning, electric, or stabbing pain [2,3,4]. Numerous pathological changes associated with diabetes development and resulting in the induction and maintenance of PDN are transmitted to the CNS via sensory and nociceptive dorsal root ganglion (DRG) neurons. These changes are ultimately expressed as alterations in the patterns of action potentials (Aps) transmitted by the primary afferents to the spinal cord. Low-voltage-activated T-type Ca^2+^ channels (T-channels), which are expressed in the primary nociceptive neurons [5,6], both regulate patterns of Aps in somas of DRG neurons [7,8,9] and play a crucial role in the processing of painful signals [10,11,12,13,14,15]. T-channels in the soma of nociceptive neurons may play a significant role in PDN maintenance and are attractive candidates for regulating nociception under diabetic conditions for the following reasons: First, upregulation of T-channels has been shown in the soma of nociceptive DRG neurons isolated from diabetic rats and mice with different painful symptoms of PDN [7,8,16,17,18,19,20,21,22]. Second, inhibition of these channels in vivo effectively reversed mechanical and thermal hyperalgesia in animal models of type 1 and type 2 diabetes [17,18], indicating that T-channel functioning in nociceptive DRG neurons is causally linked to PDN maintenance. It is well-known that T-channels regulate neuronal excitability, and we have shown that streptozotocin (STZ) diabetes-induced upregulation of T-type Ca^2+^ current (T-current) leads to lowering a threshold for AP initiation and AP bursting in the soma of capsaicin-insensitive, IB_4_-negative, low-pH-sensitive neurons (caps-) that may contribute to non-thermal nociception at later-stage diabetes [8]. However, particular molecular and cellular mechanisms that may link the increased somatic excitability of nociceptive DRG neurons with the pathological pain processing in PDN are unknown, preventing the development of targeted therapies against diabetes mellitus devoid of side effects.

Due to methodological difficulties, previous research has focused on studies of T-currents recorded at ambient temperature in the soma of DRG neurons deprived of their axons [10,11,13,14,23]. Unfortunately, these studies carried out with a reduced preparation at non-physiological temperatures have not produced conclusive results that would allow for a mechanistic explanation of PDN maintenance by upregulation of somatic T-channels.

Here, using a combination of electrophysiological and modeling approaches, we have studied how diabetes-induced functional upregulation of T-channels in the soma of primary nociceptors modifies primary afferent input to the spinal cord at normal body temperature. For that, ionic currents were recorded in nociceptive DRG neurons of control and STZ-diabetic rats and then fitted using Hodgkin–Huxley formalism. The currents were used to develop a one-compartmental model of a neuronal soma that reproduced the increased excitability of isolated nociceptive DRG neurons obtained from STZ-diabetic rats. Peripheral and central axons were further included in the model, and the propagation of APs along the axons was systematically analyzed. We have demonstrated that diabetes-induced functional upregulation of T-channels in the soma of primary nociceptors at normal body temperature resulted in somatic burst firing induced by a single AP evoked in a peripheral receptor zone. Importantly, these somatic bursts were further transmitted to the central axons, thus amplifying the primary afferent input to the spinal cord. We have suggested that this somatic T-channel-dependent amplification of primary nociceptive input to dorsal horn neurons underlies abnormal nociception in diabetic conditions.

## 2. Materials and Methods

All animal care and handling was done in accordance with the protocols of the Animal Care and Use Committee at the Bogomoletz Institute of Physiology.

### 2.1. Induction of STZ-Diabetes

Diabetes was induced by injection of STZ (50 mg/kg, i.p.) in adult male Wistar rats weighing 250–300 g. STZ was prepared in 100 mM citric acid, pH 4.2, on ice. The blood glucose level was measured on the third day with the glucometer Glucotrend (Boehringer Mannheim, Mannheim, Germany). All animals used in our experiments had long-term diabetes (>8 weeks) and a blood glucose level higher than 25 mM after the establishment of hyperglycemia and before DRG isolation.

### 2.2. DRG Neurons Preparation and Recordings

Caps- neurons were isolated from the lumbar L4–L6 DRG and electrophysiologically identified as previously described [8]. The following solutions were used in the electrophysiological experiments: An internal solution contained in mM: 120 KCl, 4 MgATP, 30 HEPES, 2.25 CaCl_2_, 10 EGTA, 10 NaCl, pH 7.4 with KOH, 296 mOsm, and an external Tyrode’s solution contained in mM: 140 NaCl, 4 KCl, 2 MgCl_2_, 2 CaCl_2_, 10 glucose, and 10 HEPES, pH 7.4 with NaOH. For measuring Ca^2+^ currents (Figure 1C), the external solution was changed to the following (in mM): 165 tetraethylammonium chloride (TEA-Cl), 10 HEPES, 2 BaCl_2_, pH 7.4 with TEA-OH, 305–315 mOsm [8]. For measuring potassium currents (K^+^ currents; Figure 1E), the same internal solution was used while the external solution was changed to (in mM): 150 choline chloride, 10 HEPES, 4 MgCl_2_, 4 KCl, pH 7.4, with Tris-base. The stimulation protocol in voltage-clamp configuration for measuring K^+^ currents consisted of depolarization steps from a holding potential of −100 mV (3.5 s) to test potentials from −60 mV to +40 mV (200 ms) with 20 mV increments. Na^+^ current was measured as described previously [8]. When applicable, recordings and mean characteristics of voltage-gated currents used for further simulations were obtained by pooling the results of current experiments and ones conducted previously [8].

### 2.3. Chemicals

Collagenase type 4, 250 U/mg, and trypsin, 204 U/mg, used for isolation of DRG neurons were from Worthington Biochemical Corporation, Lakewood, USA. Streptozotocin, used for the induction of diabetes, and other chemicals were from Merck, St. Louis, MO, USA.

### 2.4. Statistics

Statistical comparisons of data obtained from neurons of diabetic and normal rats were done using the unpaired Student’s *t*-test. One-way ANOVA was used to find out that measured values were not significantly different between animals within control and diabetic groups and could be combined into cumulative control and diabetic samples to be compared using Student’s *t*-test. Intergroup differences with *p* < 0.05 were considered to be significant. Factors that might influence the difference between neurons in control and diabetic rats, such as sex, litter, or age, were excluded by the following design. Only male siblings from several litters with approximately the same age (the difference was within 1 week) were separated into control and diabetic groups that were blindly taken care of side by side.

### 2.5. Computer Simulations

Computation experiments were performed on models of two types (Figure 2) developed in the NEURON simulation environment [24]: (1) a single-compartment model of isolated soma; (2) a multi-compartmental model of nociceptive DRG neuron composed of the soma and pseudo-unipolar process (trunk) bifurcating to form a so-called T-junction with central and peripheral branches (axons). All structural elements were non-myelinated cylinder-shaped and characterized by lengths *L* and diameters *D* in correspondence with the morphology of the vertebrate non-myelinated nociceptive primary afferent DRG neuron [25]. The soma had *L* = 15 μm and *D* = 20 μm trunk *L* = 50 µm and *D* = 0.4 µm, central and peripheral branches had *L* = 10,000 µm and 20,000 µm, respectively, and *D* = 0.5 µm. The peripheral branch was terminated with a receptor zone (*L* = 100 µm; *D* = 0.5 µm).

In both models, the membrane was populated by ion channels characteristic of the prototype cells. Partial ion currents per unit membrane area (mA/cm^2^) were described by the Hodgkin–Huxley-type equations (Hodgkin and Huxley, 1952), with parameters adjusted to fit experimental recordings when available (see below). In the below equations, *V* is the membrane potential; *G_i_* and *E_i_* are respectively the maximum conductivity per unit membrane area and reversal potential for the i-th partial current (*i* = *Na* (TTX-sensitive Na^+^ current), *K* (fast non-inactivating delayed rectifier K^+^ current), CaT (T-type Ca^2+^ current), *CaL* (L-type Ca^2+^ current), Leak); *q*10 = 3^((t° − 23)/10)^ is the dimensionless factor of dependence of kinetic rates on temperature t° in degrees Celsius. CaT- and CaL-currents were intentionally included only in the soma to study the role of the soma in T-channel upregulation in diabetic neuropathy.

The partial currents were described by the following equations:

Fast inactivating TTX-sensitive sodium current (*Na*):$$J_{Na}=G_{Na}\cdot m^{3}\cdot h\cdot \left(V-E_{Na}\right);$$
GNa=0.04 Scm2; ENa=50 mV;
$$\frac{dm}{dt}=\frac{\left(m_{\infty }-m\right)}{\tau_{m}};\;\tau_{m}=\frac{1}{q_{10}\cdot \left(\alpha_{m}\left(V\right)+\beta_{m}\left(V\right)\right)};\;m_{\infty }=\alpha_{m}\left(V\right)\cdot \tau_{m};$$
$$\alpha_{m}\left(V\right)=\frac{-0.1\cdot \left(V+40\right)}{exp\left(\frac{-\left(V+40\right)}{10}\right)-1};$$
$$\beta_{m}\left(V\right)=4\cdot exp\left(\frac{-\left(V+65\right)}{15}\right);$$
$$\frac{dh}{dt}=\frac{\left(h_{\infty }-h\right)}{\tau_{h}};\;\tau_{h}=\frac{1}{q_{10}\cdot \left(\alpha_{h}\left(V\right)+\beta_{h}\left(V\right)\right)};\;h_{\infty }=\alpha_{h}\left(V\right)\cdot \tau_{h};$$
$$\alpha_{h}\left(V\right)=0.07\cdot exp\left(\frac{-\left(V+65\right)}{20}\right);$$
$$\beta_{h}\left(V\right)=\frac{1}{exp\left(\frac{-\left(V+35\right)}{10}\right)+1},$$
where *m* and *h* are kinetic variables of activation and inactivation, respectively; *m*_∞_ and *h*_∞_ are steady-state values of *m* and *h*; *α_x_*, *β_x_*, and *τ_x_* are, respectively, the forward and backward rate constants and time constants of variables *x* = *m*, *h*.

Fast non-inactivating delayed rectifier potassium current (*K*):$$J_{K}=G_{K}\cdot n^{4}\cdot \left(V-E_{K}\right);$$
GK=0.008 Scm2; EK=−77 mV;
$$\frac{dn}{dt}=\frac{\left(n_{\infty }-n\right)}{\tau_{n}};\;\tau_{n}=\frac{1}{0.5\cdot q_{10}\cdot \left(\alpha_{n}\left(V\right)+\beta_{n}\left(V\right)\right)};\;n_{\infty }=\alpha_{n}\left(V\right)\cdot \tau_{n};$$
$$\alpha_{n}\left(V\right)=\frac{-0.01\cdot \left(V+55\right)}{exp\left(\frac{-\left(V+55\right)}{5}\right)-1};$$
$$\beta_{n}\left(V\right)=0.125\cdot exp\left(\frac{-\left(V+65\right)}{80}\right),$$
where *n* is the kinetic variable of activation; *n*_∞_ is the steady-state value of *n*; *α_n_*, *β_n_*, and *τ_n_* are, respectively, the forward and backward rate constants and time constant.

T-current (*CaT*):$$J_{CaT}=G_{CaT}\cdot m^{2}\cdot h\cdot \left(V-E_{CaT}\right);$$
GCaT=0.0001 Scm2; ECaT=120 mV;
$$\frac{dm}{dt}=\frac{\left(m_{\infty }-m\right)}{\tau_{m}};\;\tau_{m}=1.44+\frac{1}{q_{10}\cdot \left(\alpha_{m}\left(V\right)+\beta_{m}\left(V\right)\right)};\;m_{\infty }=\alpha_{m}\left(V\right)\cdot \tau_{m};$$
$$\alpha_{m}\left(V\right)=exp\left(\frac{V+51}{5.67}\right);$$
$$\beta_{m}\left(V\right)=exp\left(\frac{-\left(V+128\right)}{15.33}\right);$$
$$\frac{dh}{dt}=\frac{\left(h_{\infty }-h\right)}{\tau_{h}};\;\tau_{h}=26.67+\frac{1}{q_{10}\cdot \left(\alpha_{h}\left(V\right)+\beta_{h}\left(V\right)\right)};\;h_{\infty }=\alpha_{h}\left(V\right)\cdot \tau_{h};$$
$$\alpha_{h}\left(V\right)=exp\left(\frac{V+41}{4.32}\right);$$
$$\beta_{h}\left(V\right)=exp\left(\frac{-\left(V+423.33\right)}{53.33}\right),$$
where *m* and *h* are kinetic variables of activation and inactivation, respectively; *m*_∞_ and *h*_∞_ are steady-state values of *m* and *h*; *α_x_*, *β_x_*, and *τ_x_* are, respectively, the forward and backward rate constants and time constants of variables *x* = *m*, *h*.

L-current (*CaL*):$$J_{CaL}=G_{CaL}\cdot m^{2}\cdot h\cdot \left(V-E_{CaL}\right);$$
GCaL=0.00025 Scm2; ECaL=120 mV;
$$\frac{dm}{dt}=\frac{\left(m_{\infty }-m\right)}{\tau_{m}};\;\tau_{m}=\frac{2.02}{q_{10}\cdot \left(\alpha_{m}\left(V\right)+\beta_{m}\left(V\right)\right)};\;m_{\infty }=\alpha_{m}\left(V\right)\cdot \tau_{m};$$
$$\alpha_{m}\left(V\right)=\frac{-0.055\cdot \left(V+32\right)}{exp\left(\frac{-\left(V+32\right)}{3.8}\right)-1};$$
$$\beta_{m}\left(V\right)=0.94\cdot exp\left(\frac{-\left(V+80\right)}{17}\right);$$
$$\frac{dh}{dt}=\frac{\left(h_{\infty }-h\right)}{\tau_{h}};\;\tau_{h}=\frac{2.26}{q_{10}\cdot \left(\alpha_{h}\left(V\right)+\beta_{h}\left(V\right)\right)};\;h_{\infty }=\alpha_{h}\left(V\right)\cdot \tau_{h};$$
$$\alpha_{h}\left(V\right)=0.000457\cdot exp\left(\frac{-\left(V+18\right)}{50}\right);$$
$$\beta_{h}\left(V\right)=\frac{0.0065}{exp\left(\frac{-\left(V+20\right)}{28}\right)+1}.$$

Passive leak current (*Leak*):$$J_{Leak}=G_{Leak}\cdot \left(V-E_{Leak}\right);$$
GLeak=0.04 mScm2; ELeak=−80 mV.

The spatial channel distribution is shown in Figure 2. The membrane of the soma (both isolated and as a cell part) contained channels conducting *J_Na_*, *J_K_*, *J_CaT_*, *J_CaL_*, and *J_Leak_* currents. The membranes of the trunk, peripheral, and central axons contained the same but different *J_CaT_* and *J_CaL_* currents. In the single-compartment model, the membrane potential changes due to the total membrane current being the sum of the partial currents $J_{m}=\sum J_{i}$ were described as:$$C_{m}\cdot \frac{dV}{dt}=J_{m}+J_{ext}=\sum J_{i}+J_{ext}$$
where Cm=1µFcm2 is the membrane specific capacitance and $J_{ext}$ is the current from external source. The electrical behavior of the multi-compartmental model obeyed the cable equation:$$\frac{1}{r_{i}}\cdot \frac{d^{2}E}{dx^{2}}=c_{m}\cdot \frac{dV}{dt}-j_{m}$$
where *x* is the longitudinal (axial) coordinate, $c_{m}=C_{m}\cdot \pi \cdot D$ and $j_{m}=J_{m}\cdot \pi \cdot D$ are, respectively, the membrane capacitance and total current per unit length along *x*, ri=4Raπ·D2 is the cytoplasmic resistance per unit length (the cytoplasmic resistivity $R_{a}=70\;\mathrm{Ohm}\cdot \mathrm{cm}$ was homogeneous), and *D* is the diameter of the compartment. Standard equations of axial current conservation describe electrical coupling at branching points and diameter heterogeneities (e.g., between the soma and trunk).

The model can be obtained upon request.

Standard voltage-clamp protocols were used in computational experiments targeted at the validation of the simulated ion currents (Figure 1) by comparison with current traces recorded in vitro from the prototype cells (Figure 1, left column). Standard current-clamp stimulation protocols were used to evoke APs by applying depolarizing currents either to the isolated soma (restricted somatic model) or to the receptor zone (grey segment in Figure 2B) of the peripheral process.

**Figure 1 biomolecules-13-01320-f001:**
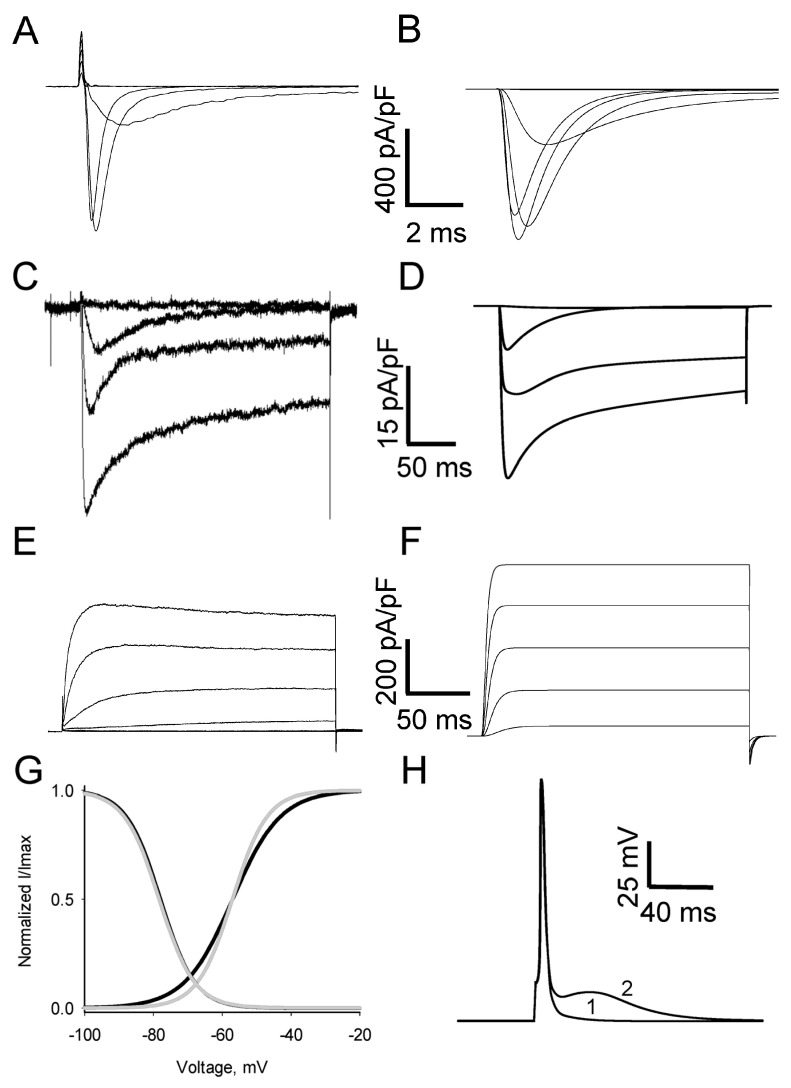
Development of model of caps- neuron. Experimental (**A**,**C**,**E**) and respective simulated (**B**,**D**,**F**) ion currents involved in generation of APs in caps- neurons. Representative traces of sodium (**A**), potassium (**C**), and calcium (**E**) currents evoked in different isolated caps- neurons by depolarizing voltage steps from a holding potential of −100 mV to testing potentials −40, −20, 0, and +15 mV (**A**), −40 to +40 with 20 mV increment (**C**), and −70 to −10 with 20 mV increment (**E**). Different intra- and extracellular solutions were used for the isolation of respective currents. Simulated traces of Na (**B**), K (**D**), and total calcium (**F**) currents obtained for the same holding and testing potentials as experimental traces. Parameters of simulated currents were fitted according to current-voltage relationships of respective experimental currents. The total Ca current (**E**) consisted of high and low threshold components and was well-fitted by a sum of CaT and CaL components (**F**). (**G**) Steady-state activation and inactivation curves of T-currents obtained in the prototype and simulated DRG neurons (grey traces from [26]) and black traces, respectively). (**H**) APs in a restricted somatic model of caps- neuron (Figure 2A) including Na-, K-, and CaL- and CaT-currents fitted as described in the Methods and shown in (**B**,**D**,**F**). APs were simulated (*V_hold_* = −80 mV) in response to 1 ms threshold current injections. The conductance of the T-current was 0 and 0.16 mS/cm^2^ for traces 1 and 2, respectively. Note a substantial afterdepolarizing potential, ADP, induced by activation of T-current.

**Figure 2 biomolecules-13-01320-f002:**
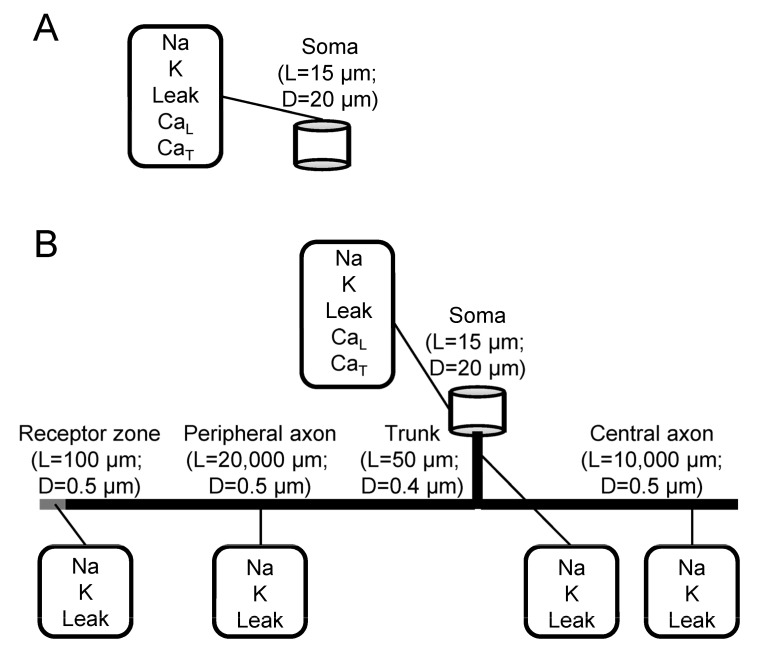
Morphological and biophysical parameters of DRG neuron models. Cylinder-shaped (lengths L, diameters D) single-compartment soma (**A**) and multicompartment neuron (**B**) models include Hodgkin-Huxley-type fast inactivating sodium (Na), non-inactivating potassium (K), and passive leakage (Leak) conductances homogeneously distributed over the cell membrane, including trunk, peripheral, and central axons. In both models, the soma contained both T-type (CaT) and L-type (CaL) Ca^2+^ conductances.

The validity of the models was estimated by comparison of the simulated and experimentally recorded currents and membrane potentials and was achieved by fitting the corresponding parameters.

## 3. Results

### 3.1. Development of Single-Compartment Model of Nociceptive DRG Neurons

The caps- nociceptive DRG neurons were routinely selected based on their size, presence of fast-inactivating T-current, and K^+^ current characteristics, as previously described [8]. In order to obtain parameters of voltage-gated channels expressed in these neurons in control conditions, we recorded Na^+^, K^+^, and Ca^2+^ currents evoked in these cells in a voltage clamp configuration of whole-cell patch clamp. Depolarization steps (200 ms) from −100 mV to different testing membrane potentials were applied to the cells in the necessary combinations of extracellular and intracellular solutions (see Methods for details) in order to separately record Na^+^, K^+^, and Ca^2+^ currents (Figure 1A,C,E). The duration of depolarization steps was set at 200 ms since AP bursts in the caps- neurons were mainly observed in vitro during the first 100 ms after the onset of depolarization [8]. Thus, currents that may contribute to AP bursts should have been recorded using the given duration of depolarization.

Voltage-gated Na^+^ and Ca^2+^ currents were recorded from a group of caps- neurons (n = 5 for Na^+^ and n = 5 for Ca^2+^ currents, Figure 1A,E). Current-voltage relationships for these currents were not significantly different from ones observed in our previous experiments with caps- neurons [8] and, therefore, characteristics obtained in this work for Na^+^ and Ca^2+^ currents were used to guide simulations.

Both fast inactivating TTX-sensitive and TTX-resistant Na^+^ currents were recorded in caps- neurons. We have previously shown that the TTX-resistant Na^+^ current is negligible compared to the TTX-sensitive one in the somas of these neurons [8]. Therefore, only fast-inactivating TTX-sensitive Na^+^ current (Figure 1A) was included in the neuronal model used in this work. Model parameters of TTX-sensitive Na^+^ current were fitted to closely reproduce (in a single-compartment soma model (Figure 2A)) the current–voltage relationship of experimentally recorded currents (compare Figure 1A and Figure 1B).

Potassium currents (K^+^ current) in the caps- neurons had noticeable activation at −20 mV, slow onset (*τ_onset_* = 5.3 ± 0.2 ms, n = 4), and slow inactivation (*τ_inact_* fells within a range of 75–158 ms with a mean value of 127 ± 26 ms, n = 4, Figure 1C). These properties of K^+^ currents suggest that non-inactivating delayed rectifier potassium channels are the major class of potassium channels mediating K^+^ current in the caps- DRG neurons [27]. Thus, the Hodgkin-Huxley model [28] of non-inactivating delayed rectifier potassium current fitted against an experimentally recorded current-voltage relationship was used for further model simulations. Examples of experimentally recorded and simulated (in a single-compartment soma model (Figure 2A)) K^+^ currents are shown in Figure 1 (Figure 1C,D).

A total Ca^2+^ current was recorded in a wide range of testing membrane potentials (voltage steps from a holding potential of −100 mV to testing potentials in a range from −80 to 0 mV; Figure 1E). This current represented a sum of low and high threshold Ca^2+^ currents [8,26] and was fitted in a single-compartment soma model (Figure 2A) by a sum of T- and L-currents (Figure 1F). Model parameters for T- and L-currents were fitted to reproduce (in a single-compartment soma model (Figure 2A)) current–voltage relationships for both peaks and steady-state levels (last 10 ms of the test step) of the experimentally recorded total Ca^2+^ currents and steady-state activation and inactivation of T-current (Figure 1G).

It has been shown that diabetes-induced changes in steady-state characteristics of T-currents increase somatic excitability [7,21,22] in nociceptive DRG neurons. Thus, it was important to include proper parameters for both steady-state activation and inactivation of these channels in our models. Unfortunately, a substantial contribution from high threshold Ca^2+^currents at high testing membrane potentials (above −40 mV) prevented us from precisely estimating amplitudes of T-current. Therefore, data for the steady-state activation and inactivation of T-channels obtained for sensory neurons [26] were used for initial simulations (Figure 1G). These data were mildly fitted to reproduce (Figure 1F) the experimentally recorded total Ca^2+^ current (Figure 1E). Note just a minor difference in the slope of steady-state activation and almost equal values of half activation and inactivation for the initial [26] and used in the model (Figure 1G) steady-state relationships.

Aps simulated in a model containing a set of the above-described channels in response to a short (1 ms) threshold current stimulation are shown in Figure 1H. Note that the inclusion of T-channels in the model resulted in a prominent afterdepolarization potential (ADP) experimentally observed in the caps- neurons [8,29].

Thus, we established a main set of voltage-gated currents activated in caps- neurons in response to membrane depolarization. Based on the experimentally obtained characteristics of these currents, we acquired the parameters necessary for their simulation. Using the parameters in a single-compartment soma model of caps- neurons, we reproduced APs observed in the caps- neurons in current clamp recordings.

### 3.2. A Single-Compartment Model Reproduces Increased Excitability of Nociceptive DRG Neurons under Diabetic Conditions

#### 3.2.1. Role of T-Type Ca^2+^ Current Density

The mean value of amplitudes of T-current density at a depolarization step from −100 mV to −50 mV in STZ-diabetes was significantly larger compared to the control conditions (6.7 ± 1.8 pA/pF for the control, n = 32, and 11.6 ± 1.7 pA/pF for the diabetes, n = 17, *p* < 0.001), while their kinetics were similar (Figure 3A). These results were previously interpreted as a diabetes-induced increase in the density of T-channels [8]. Besides, T-currents recorded from caps- neurons of control and STZ-diabetic rats revealed substantial cell-to-cell variability (Figure 3A). Thus, the T-current conductance varied in simulation experiments to reproduce a range of the values of T-current density recorded in the control and diabetic conditions. T-currents simulated in a single-compartment soma model of caps- neuron (Figure 2A) having a T-channel conductance of 0.14 (black trace), 0.20 (red trace), and 0.36 (pink trace) mS/cm^2^ fairly reproduced experimentally recorded currents obtained from the caps- neurons of control and diabetic rats (Figure 3A,B).

It has also been previously shown that TTX-sensitive and TTX-resistant Na^+^ currents, as well as high-threshold Ca^2+^ currents, are not changed in caps- neurons in STZ-diabetic conditions [8]. Here we also studied if K-currents were modulated by the diabetic conditions. For that, these currents were recorded in caps- neurons of control and STZ-diabetic rats using voltage steps (duration of 200 ms) from a holding potential of −100 to −60 through +40 mV in 20 mV increments. Averaged values of these currents for the last 10 ms of each step were calculated and compared for the neurons of control and diabetic rats. At a depolarization step from −100 to +40 mV, the mean value of K-current density in control neurons was 485 ± 32 pA/pF (n = 19), and this value was not significantly different from the corresponding value in neurons of STZ-diabetic rats, 491 ± 39 pA/pF (n = 17; *p* > 0.05). Thus, we concluded that T-current was the only voltage-gated current activated by fast depolarization that was modulated in caps- neurons under long-term diabetic conditions. In further simulation experiments, all but T-current parameters were set to the same values for control and diabetic conditions.

An AP simulated in the one-compartment somatic model having a conductance density of T-current (*G_CaT_*) of 0.14 mS/cm^2^ is shown in Figure 3C (black trace). The AP was elicited by a 1 ms threshold depolarizing current from a holding potential of −80 mV. This stimulation current caused rapid activation of T- and Na-currents, which led to rapid depolarization of the plasma membrane and the generation of a single AP (Figure 3C,D). The AP repolarization was caused by a combination of Na-current inactivation and activation of K-current (Figure 3C). If Na-current was removed from the model, no AP was generated. Removing the K-current extended the duration of AP from 1.4 to about 4.0 ms. The presence of T- current in the model having *G_CaT_* = 0.14 mS/cm^2^, which was specific for the neurons of control rats, resulted in afterdepolarization, ADP, lasting tens of ms after AP termination (Figure 3B, black trace). Increasing *G_CaT_* to 0.20 mS/cm^2^, used to reproduce the diabetic conditions, led to ADP enhancement and achieved a threshold for the generation of a second AP (Figure 3B, red trace). A subsequent increase of *G_CaT_* to 0.32 mS/cm^2^, also modeling diabetic conditions, resulted in the generation of a burst of three spikes (Figure 3C, pink trace). We have previously shown that a proportion of caps- neurons in STZ-diabetic rats generated bursts of 2–3 spikes in response to a threshold pulse stimulation that was a result of diabetes-induced upregulation of T-channels [8]. Thus, the single-compartment model also quantitatively reproduced the bursting of caps- neuron in ‘diabetic’ conditions by means of an increase in T-current conductance density.

Next, we analyzed the contribution of different voltage-gated conductances in spike generation at *G_CaT_* = 0.14 mS/cm^2^, resulting in ADP after the first spike, and at *G_CaT_* = 0.20 mS/cm^2^, producing the second spike (Figure 3C,D). Figure 3D clearly shows that Na, K, and L-type Ca^2+^ currents rapidly deactivated after the first stimulus-induced spike in both ‘normal’ (*G_CaT_* = 0.14 mS/cm^2^, black traces) and ‘diabetic’ (*G_CaT_* = 0.20 mS/cm^2^, red traces) conditions, while the inward T-current lasted longer because of its slow deactivation. It is this latter current that produces a depolarizing ramp in the neuron membrane potential, leading to ADP in ‘normal’ conditions and achieving a threshold for the generation of a second spike in ‘diabetic’ ones (Figure 3D, top traces).

Values of T-current density in the caps- neurons in control conditions were relatively high compared to ones observed in other small DRG neurons [21,22]. However, T-channels are expressed even at higher densities in medium- and large-sized DRG neurons. In particular, Nelson et al. reported [30] that there is a subpopulation of medium-sized DRG nociceptors that express a density of T-currents that is almost one order of magnitude higher (~50 pA/pF) than the one we observed in caps- nociceptors (~7 pA/pF; Figure 3A). The T-current density is further increased in pathological conditions in both the caps- neurons [8] and the medium-sized neurons [30,31]. Such high levels of expression can potentially result in spontaneous AP generation, which may underlie the spontaneous pain observed in many pathologies, including diabetic neuropathy. Therefore, we further checked if an increased density of T-currents that was observed in the nociceptors might result in their spontaneous activity. T-channel conductance density to values observed in medium-sized neurons (1.4 mS/cm^2^) resulted in spontaneous bursting (Figure 3E), while a further increase to 2.2 mS/cm^2^ produced regular firing (Figure 3F).

Thus, we have demonstrated that the somatic model of a nociceptive DRG neuron, populated by a set of voltage-gated channels, reproduced the enhanced excitability observed in these neurons at ambient temperature under diabetic conditions. This enhanced excitability shown in simulation experiments was only due to the increased T-channel conductance; it is important that the conductance values used in the model resulted in densities of simulated T-currents that were similar to those observed in conditions of diabetic-induced upregulation of the Ca_V_3.2 subtype of T-channels [8].

#### 3.2.2. Role of Steady-State Inactivation

It has been shown that in small capsaicin-sensitive IB4-positive nociceptive DRG (caps+) neurons having sizes similar to caps- neurons and expressing a similar level of T-channel expression, experimental diabetes resulted in a depolarizing shift of steady-state inactivation of T-current. It has been suggested that the observed shift in inactivation may contribute to PDN maintenance [21,22]. We hypothesized that this shift may increase nociceptive neuron excitability and, therefore, simulated neuronal electrical activity in the single-compartment soma model (Figure 2A), introducing the shift of steady-state inactivation of T-current in the depolarizing direction by 8 mV (Figure 4A) to mimic the development of diabetes. Expectedly, such a shift at current-clamp conditions resulted in an increased ADP and, correspondingly, the generation of a second spike (Figure 4B, top). Simulations in voltage-clamp conditions showed that increased ADP and the generation of the second spike are due to a substantial increase in T-current at the membrane potential close to the threshold of AP generation (Figure 4B, bottom). Thus, both phenomena observed in DRG neurons of diabetic rats—an increased T-current density and a depolarizing shift of steady-state inactivation—result in an enhancement of excitability of the modeled nociceptive DRG neurons.

The values of conductance and *V*_1/2_ were systematically varied in the simulations in order to demonstrate how changes in these parameters under control and in diabetic conditions may affect the excitability of nociceptive neurons. A phase diagram shown in Figure 4C demonstrates how changes in density and steady-state inactivation of T-current in diabetic conditions may lead to the somatic generation of several rather than one AP in response to the threshold current stimulation (Figure 4C). The diagram clearly shows a set of parameters of T-currents observed in small nociceptive DRG neurons in diabetic conditions, emphasizing the possibility of a substantial increase in the number of somatic Aps under diabetes due to T-channel upregulation. Thus, we have shown that diabetic-induced changes in the biophysical properties of T-channels do result in the excitability of nociceptive DRG neurons.

#### 3.2.3. Role of Temperature

Patch-clamp studies of the electrophysiological properties of DRG neurons have been almost exclusively performed with isolated neurons at room temperature. Although it is generally accepted that ionic currents are temperature dependent and this temperature-dependence is even included in simulation environments, e.g., the neuron used in this work, no modeling related to a better understanding of how temperature can influence diabetic-induced T-channel-dependent increases in excitability of nociceptive DRG neurons has been carried out. In the following simulations, the dimensionless factor of dependence of kinetic rates on temperature t° in degrees Celsius, *q*10, was set as *q*10 = 3^((t° − 23)/10)^.

First, we tested if the excitability of cap- neurons can be increased at 36 °C compared to 22 °C in the case of mild diabetic-induced upregulation of the conductance density of T-current, *G_CaT_*. For that, we choose *G_CaT_* equal to 0.16 mS/cm^2^ and the voltage of half-maximal steady-state inactivation (*V*_1/2_) equal to −78 mV. In spite of an increase in the conductance density compared to normal conditions (*G_CaT_* = 0.10–0.14 mS/cm^2^), the simulation with these parameters at 22 °C still produced just one AP. However, changing the temperature to 36 °C resulted in an enhancement of excitability and the generation of three APs in response to the threshold stimulation (Figure 5A). At the same time, at 36 °C and *G_CaT_* = 0.12 mS/cm^2^, only one AP was generated, indicating that mild experimentally observed changes in T-currents may produce substantial enhancement of the excitability of nociceptive neurons.

The numbers of APs generated at different values of conductance densities of T-current, *G_CaT_*, and temperature are shown on a respective phase diagram (Figure 5D). It is clear from Figure 5D that the phase curves dividing the phase space into regions with certain numbers of APs are closer to each other at 36 °C than at 22 °C. It means that the same changes in *G_CaT_* should result in a more substantial modulation of neuronal excitability. Indeed, an increase in *G_CaT_* from 0.10 mS/cm^2^ (normal conditions) to 0.19 mS/cm^2^ (diabetic conditions) leads to a dramatic enhancement of excitability from 1 to 4 APs in response to a threshold stimulation (Figure 5B). Temperature also modifies the patterns of AP bursts. In general, an increase in temperature from 22 °C to 36 °C shortens the burst duration and substantially increases the firing rate of APs within the burst (Figure 5C). In all examples shown above, the integral values and waveforms of T-current were substantially changed in accordance with temperature, underlying the main component of changes in neuronal excitability (Figure 5A–C; bottom rows).

The observed temperature-dependent changes in patterns of AP bursts can also be attributed to temperature-dependent modulation of the Na-current. Next, we checked the temperature-dependent changes in CaT-current and Na-currents to better understand their relative contributions to patterns of AP bursts. As it is clear from Figure 5E, the voltage dependence of activation time constants for CaT and Na conductance is strongly affected by temperature. The same is valid for other kinetic constants of these currents. It is interesting that it is the activation constant for Na conductance that is strongly affected by temperature in the range of membrane potentials close to the threshold of AP generation (−45–−35 mV; Figure 5E). In accordance with that, simulations of CaT-current and Na-currents in voltage clamp mode demonstrate substantially larger temperature-dependent changes in the activation kinetics of Na-current compared to CaT-current (Figure 5F,G). At the same time, the amplitudes of both currents were just slightly affected by changes in temperature from 22 °C to 36 °C. The deactivation kinetics of both currents were substantially faster at 36 °C compared to 22 °C. Altogether, the increase in temperature results in substantially faster Na-current (Figure 5G) and faster deactivation of CaT-current (Figure 5F). It can be predicted from these results that a larger number of APs can be generated during ADP development due to T-channel activation induced by the first spike. Indeed, simulations carried out at 36 °C with the activation time constants of Na-current set to values at 22 °C and 36 °C demonstrate that in the former case, two APs are generated, while in the latter, three APs are observed (compare the red traces in Figure 5G and Figure 5A). If the ADP induced by T-channel activation in a simulation of neuronal activity at 36 °C is prolonged using kinetic constants for CaT-current obtained at 22 °C, then the number of generated APs is increased to nine instead of three (compare the red traces in Figure 5H and Figure 5A). Thus, at higher temperatures, faster Na-current results in an increased firing rate of APs during the ADP, while the duration of the ADP, mainly defined by CaT-current kinetics, is reduced. However, during this shorter ADP, more APs have a chance to be generated, leading to increased excitability of nociceptive DRG neurons at 36 °C compared to those at 22 °C. This suggests that the excitability of nociceptive DRG neurons at normal body temperature is much more sensitive to an increase in T-channel density than at room temperature. In other words, the mild diabetic-induced increase in the excitability of nociceptive neurons, including caps- observed in electrophysiological experiments at room temperature [8] is likely substantially enhanced at the normal animal’s body temperature.

### 3.3. A Comprehensive Model of Nociceptive DRG Neurons Demonstrates Somatic Amplification of Peripheral Input under Diabetic Conditions

To study the role of somatically localized T-channels in the AP output of nociceptive DRG neurons, we created a computational model of a C-DRG neuron having an unmyelinated trunk and peripheral and central axons (Figure 2B). In the model, under control conditions and normal body temperature (36 °C), an AP evoked in a receptor zone of the peripheral process propagated along the axon, invaded the soma, and simultaneously further propagated to the end of the central axon (Figure 6A). Increased density of somatic T-current, observed in DRG neurons under diabetic conditions [8], transformed a single AP invading the soma into a burst of several APs (Figure 6B). It is interesting to note that the interspike interval in the burst was longer than 15 ms. It allowed Na- and K-channels localized in the trunk and inactivated after the first AP invaded from the periphery to recover and transmit the second and third APs in the burst to the conducting unmyelinated axon. Two extra spikes were added to the first one traveling toward the end of the central axon, leading to a three-fold increase in the number of APs reaching the spinal cord. Besides, these extra spikes also travel antidromically in the peripheral axon (Figure 6B).

A shift in steady-state inactivation of T-current observed in diabetic conditions [7,21,22] also led to a somatic amplification in the number of APs reaching the end of the central axon (Figure 6C). Simultaneous changes in steady-state inactivation and current density produced a large ADP with a barrage of spikelets on its top in the soma of the neuron (Figure 6C). It is important that these small somatic spikelets develop into full-shaped APs when they reach the end of the central axon. Thus, one peripheral AP was somatically amplified to a burst with five APs reaching the spinal cord (Figure 6D).

Then, in a series of simulation experiments, we modeled the dependence of the somatic amplification of neuronal AP output on T-channel density (Figure 6E) and steady-state inactivation (Figure 6F). These simulations demonstrate the profound and steep dependence of somatic amplification on the above parameters at a temperature of 36 °C. For example, a 1.5–2.0 fold increase in T-channel density that is in line with an experimentally observed diabetes-induced increase [7,8,21,22] led to a 3–5-fold amplification of the AP output of nociceptive DRG neurons (Figure 6E). It is also important to note that the steepness of the AP output dependence on the studied parameters was strongly dependent on the temperature (compare the black and red traces in Figure 6E,F). It is also clear that at higher densities of T-channels, when the soma of nociceptive neurons can generate spontaneous bursts or regular firing (Figure 3E,F), these somatic APs are also transmitted to the main axon, resulting in a substantial output of nociceptive neurons even without a peripheral input. Thus far, in simulation experiments, we have demonstrated that in diabetic conditions and at normal body temperature, upregulation of T-channels in the soma of nociceptive DRG neurons results in a strong enhancement of the AP output of these neurons.

## 4. Discussion

This work is the first attempt to extend the functional studies of molecular and cellular mechanisms of PDN from the soma of dorsal root ganglion neurons to their central and peripheral axons by means of a combination of electrophysiological studies and computer modeling. Here we have conducted a set of experiments focused on modeling enhanced AP output in the central axons of nociceptive DRG neurons induced at normal body temperature by diabetic-induced upregulation of T-channels in their soma. A particular appeal of the present work lies in the evidence that diabetes-induced upregulation of the Ca_V_3.2 isoform of the T-channels in the soma of nociceptive DRG neurons enhances the AP output of these neurons, thereby offering a simple cellular mechanism that may underlie PDN.

### 4.1. Development of New Models of Nociceptive DRG Neurons

In the present work, we have, for the first time, employed detailed modeling for the functional analysis of diabetes-induced T-channel-dependent modulation of the output of primary nociceptors. Previously developed models of nociceptive DRG neurons have not considered the biophysical properties of identified subtypes of nociceptors and particular values of currents observed in these cells in normal and diabetic conditions [32,33,34,35,36,37,38,39] and, therefore, could not be used for such analysis. The newly established models are adequate for the prototype caps- and partially caps+ DRG neurons, in which T-channels are upregulated as a result of diabetes development. Initially, electrophysiologically identified voltage-gated currents expressed in caps- neuron, which may contribute to non-thermal nociception at later-stage diabetes, and used them to develop a multi-compartmental model of these neurons, including peripheral and central processes. A single-compartment somatic part of this model was fitted to reproduce currents and patterns of activity observed in the isolated caps- neurons during whole-cell electrophysiological recordings. The model has been used to address problems that are hardly resolvable because of the methodical restrictions of biological experiments. Namely, we have studied the role of somatic upregulation of T-current in the AP output in the central axons of primary nociceptive neurons at normal body temperature. The obtained results mechanistically explain how somatic T-channel upregulation increases the primary nociceptive input to the spinal cord under diabetes mellitus and warrant further modeling studies. We have only studied the somatic amplification of single peripheral spikes. However, somatically generated extra spikes produce the AP or AP bursts traveling antidromically. In the peripheral axon, they potentially collide with and occlude normal orthodromic Aps. Our simulations indicate that the time necessary for the APs to propagate from the soma to the peripheral receptor zone, together with the burst duration, is less than 100 ms. Hence, a strong amplification of peripherally evoked APs would be observed only at frequencies lower than 10 Hz. However, at higher frequencies of peripheral AP generation, extra spikes would result in complex changes in the patterns of afferent APs reaching the spinal cord, which is interesting to study.

It is well known that T-channels are expressed not only in the soma of nociceptive neurons but also in their peripheral and central axons, and their spatial distribution can be studied using immunochemical and genetic approaches [12,40]. Incorporation of data about the spatial T-channel distribution under normal and pathological conditions in biologically inspired models promotes comprehension of PDN mechanisms. In future research, it would be interesting to focus on biophysical (T-channel properties and spatial distribution) and structural (geometry of DRG soma and processes) determinants of transition from normal firing mode, i.e., simple centripetal transfer of peripherally evoked APs, to “pathological” modes when the cell multiplies the number of incoming APs or generates intrinsic sustained firing in the absence of peripheral input. The modeling results of this work also have a translational value for studies of other T-channel-associated neuropathies [41,42] or any channelopathies in DRG neurons [23]. They demonstrate that detailed models of primary nociceptive neurons populated with ion channels having characteristics of the prototype cells will allow for the pinpointing of electrical, geometrical, and structural parameters of the models that are critical for the occurrence of ‘pathological’ firing patterns in these neurons.

### 4.2. Role of Somatic Amplification in Pathological States

Recent reports have established the direct relationship between changes in pain sensation and ectopic or amplified AP activity in the sensory afferents [43,44,45,46]. The molecular and cellular mechanisms of this activity are still unclear for many pathologies, and the somatic amplification shown here might be considered one of the possible mechanisms.

It is important to emphasize that most studies of acutely dissociated nociceptors, including all of those related to the investigation of T-channels, fail to demonstrate spontaneous bursting, and only a few demonstrate evoked bursting under pathological or artificial conditions [8,30,47]. It indicates that the somas themselves are not a source of pathological AP activity, and the contribution of somatic amplification of peripherally evoked AP generation shown in this work (Figure 6) may be a mechanism explaining an amplified AP input to the spinal cord. No or low spontaneous activity has also been recorded in small- and medium-sized DRG neurons of naïve rats in intracellular recordings obtained from neurons within intact isolated L4 and L5 ganglia, although pathological changes can produce such activity [36,48,49,50]. These results directly demonstrate the somatic or near-somatic site of amplified spontaneous AP activity observed in different pathological conditions and shown in this work (Figure 3E,F). At the same time, in vivo unit recordings from C-fiber afferents demonstrate that low ongoing background AP activity does occur. For example, under controlled in vivo conditions, the incidence of nociceptor spontaneous AP activity, extracellularly recorded in their axons, has been reported at a level of 7–13% [51,52,53]. Similar to results obtained with isolated neurons and ganglia, both spinal injury and inflammation substantially increase the percentages of nociceptive C-fiber neurons with spontaneous activity [52]. It has been suggested in this and several other studies that the spontaneous activity observed in vivo in certain types of peripheral neuropathies in rats (not including diabetic ones) is mainly generated at the periphery [50,53,54]. This suggestion is directly supported by observations in which a percentage of C-neurons exhibiting spontaneous activity increased following peripheral inflammation or neuropathy, and this spontaneous firing ceased completely when the peripheral nerves were sectioned [47,52]. Unfortunately, these results do not provide an opportunity to understand whether somatic amplification is involved in the spontaneous activity of nociceptive neurons and its pathological increase since the sectioning does not allow peripheral action potentials to reach the somas of neurons to be amplified. At the same time, the results do not rule out such a possibility.

Other in vivo unit recordings from C-fibers in pathological conditions have revealed spontaneous activity generated or amplified in or near the soma [55], implying the somatic amplification suggested in this work. Ectopic somatic and axonal spontaneous activity was also observed in neurons of the chronically compressed dorsal root ganglion [49]. Importantly, that bursting evoked by a single sciatic nerve stimulation was also observed in somas of DRG neurons in elegant in vivo intracellular microelectrode recordings, thus far directly demonstrating somatic or near somatic AP amplification; the incidence of this evoked bursting was significantly increased after chronic compression of the DRG, suggesting a possible mechanism underlying tactile allodynia [56].

The above results strongly suggest that spontaneous and evoked bursting of the nociceptor is rare under normal conditions and appears or is enhanced in different pathological conditions at least partially due to the somatic amplification.

### 4.3. Relation of Results to PDN

AP activity in DRG nociceptors excites pain pathways, drives central sensitization, and results in pain, allodynia, and hyperalgesia [57]. Although it is generally accepted that T-channels, by being activated near resting membrane potential, have biophysical characteristics suited to facilitate AP generation and rhythmicity, no certain molecular and cellular mechanisms have been suggested that explain how diabetes-induced T-channel upregulation can transform peripheral input into abnormal central output of nociceptive DRG neurons.

Spontaneous pain, mechanical hyperalgesia, and tactile allodynia may persist in longer-term diabetes [58,59], suggesting sensitization of capsaicin-insensitive mechanical and/or chemical rather than thermal nociceptors. In the previous work, we demonstrated that in longer-term diabetes, capsaicin-insensitive low-pH-sensitive types of DRG neurons, caps-, show diabetes-induced upregulation of the Cav3.2 subtype of T-channels in their somas that results in ADP and the appearance of somatic double-spike bursts evoked by threshold current injections [8]. In the current work, we have determined and quantified a main set of currents expressed in this type of neuron, included these currents in a computer model, and demonstrated the appearance of somatic bursting when T-channel density was increased to values observed under STZ-diabetes (Figure 3). Moreover, an additional increase in density led to spontaneous somatic AP generation (Figure 3). Further incorporation of peripheral and central axons into the model has allowed us to demonstrate that a single peripherally induced AP propagating to the soma is amplified to several APs, which return to the axon propagating in both central and peripheral directions (Figure 6). And this somatic amplification was obtained when the T-current density was upregulated to values observed in the caps- neurons under diabetic conditions.

It is important to note that in this work we have shown that phase curves in a phase space of T-type current conductance density and temperature get closer at higher temperatures (Figure 5D). As a result, several times larger increases in the T-channel density are necessary at 22 °C (~0.08 mS/cm^2^) compared to one at 36 °C (~0.03 mS/cm^2^) in order to increase from one to three the number of APs generated by the nociceptive neuron in response to a threshold stimulation (Figure 5D). It means that the excitability of nociceptive neurons is much more sensitive to diabetes-induced upregulation of T-channels at 36 °C than at 22 °C. At the same time, the amplitudes of the T-current are only slightly changed when the temperature is increased (Figure 5F). Thus, we may suggest that the T-currents experimentally observed in normal and diabetic conditions at 22 °C [8] will be similar to those at 36 °C. Taking it into account, we may conclude that a diabetes-induced increase in the T-current conductance density of ~50% (0.05–0.8 mS/cm^2^) will lead to the somatic amplification of peripheral input in the majority of caps- neurons. Thus, the particular set of voltage-gated channels expressed in the caps- neurons, their levels of expression, and their temperature determine the ability of this type of neuron to somatically amplify the peripheral input. It has been previously shown that acutely dissociated DRG cells accurately reflect the population of voltage-gated Ca^2+^ channels expressed in intact DRGs [30]. Taking it into account, we have concluded that the somatic amplification of peripheral input is most probably present in caps- nociceptors in longer-term diabetes. This amplification may certainly contribute to mechanical hyperalgesia experienced by STZ-diabetic animals and diabetic patients in the longer term since the number of APs reaching the central terminals of caps- afferents is manifold increased compared with ones originated at the periphery. Moreover, AP bursts to which single APs are converted by somatic amplification are very effective in producing dorsal horn neuronal plasticity [60] and supporting chronic pain states [61,62].

Previous studies have shown that thermal hyperalgesia/hypoalgesia in STZ-diabetic rats correlates with T-channel upregulation/downregulation in the capsaicin-sensitive IB4-positive nociceptive DRG, caps+, neurons [21]. This upregulation was mainly produced by a shift in the steady-state inactivation of the channels (Figure 4). We have modeled this upregulation and demonstrated that the shift also produces the somatic amplification of peripheral input in caps+ neurons (Figure 4 and Figure 6), analogous to the one observed in caps- neurons. Although we have not accurately modeled a set of currents expressed in the caps+ neurons, the obtained simulations indicate that somatic amplification may also be expressed in this type of nociceptors at early stages of STZ-induced diabetes, contributing to the development and maintenance of PDN. It is interesting to note that the opposite shifts in inactivation of the sodium conductances observed in acutely dissociated DRG nociceptive neurons of mice with genetic ablation of Fhf genes resulted in inhibition of neuronal AP output shown in a modeling study and the heat nociception deficit [37]. Although changes in the inactivation introduced in this model were both somatic and axonal, this work also directly demonstrates a mechanistic explanation of how a peripherally generated AP may be modulated due to channelopathies and emphasizes the need to use comprehensive models of DRG neurons for their studies.

The T-channels are also a critical regulator of the cellular excitability of nociceptive medium-sized DRG neurons termed “T-rich” cells, which are sensitive to capsaicin and are IB4-positive [30]. Prominent T-currents in “T-rich” cells isolated from the DRG generate visible ADPs and high-frequency burst firing. The density of T-currents in “T-rich” cells is several fold higher compared to those observed in caps+ and caps- DRG neurons. This density is in a range of values that may induce spontaneous burst firing and even ectopic autonomous AP generation, as shown in Figure 3. According to their soma size, “T-rich” cells are likely to be Aδ myelinated afferents. Upon diabetes-induced T-channel upregulation and the following somatic amplification of peripheral input demonstrated in this work, they can participate in increased responses to innocuous stimuli, contributing to tactile allodynia. They may also potentially generate spontaneous bursts or tonic AP activity, resulting in excessive activity independent of peripheral input and thus underlying the maintenance of spontaneous pain.

We expect that future research will allow us to directly experimentally validate whether the maintenance of painful diabetic neuropathy is contributed by facilitated or sustained firing in the axons of certain types of primary nociceptive neurons that occur due to diabetes-induced upregulation of somatic T-type Ca^2+^ channels.

## 5. Conclusions

Altogether, the somatic T-channel-dependent amplification of peripheral APs demonstrated in this work may strongly contribute to the enhancement of the primary nociceptive input to the spinal dorsal horn neurons and thus underlie the abnormal nociception at different stages of diabetes development.

## Figures and Tables

**Figure 3 biomolecules-13-01320-f003:**
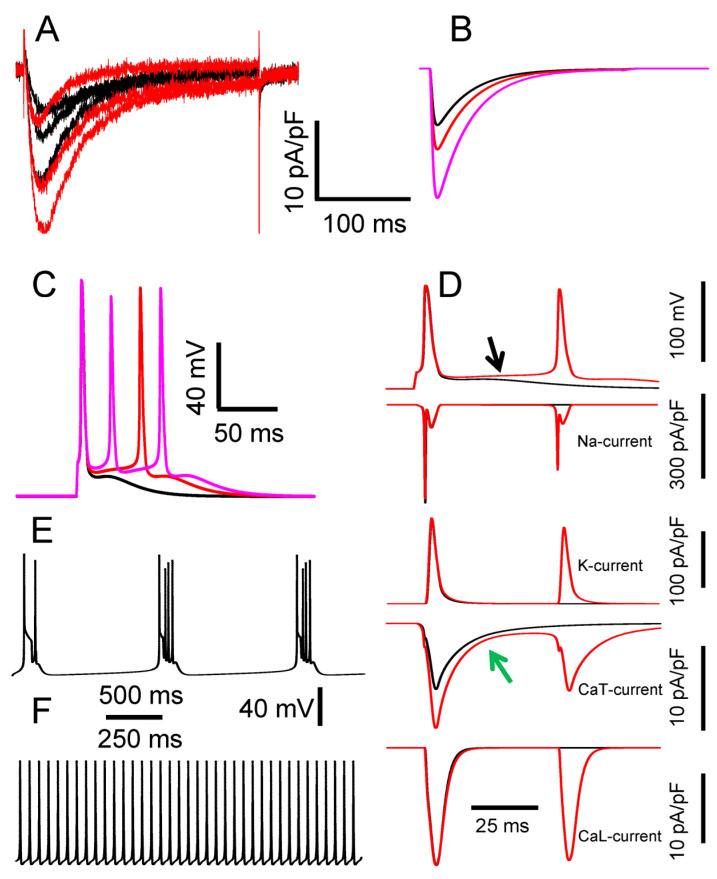
Modeling of increased excitability of caps- neurons under diabetic conditions due to upregulation of T-current density. (**A**) Representative mean T-currents recorded from caps- neurons in control (black traces; n = 3) and STZ-diabetic (red traces; n = 3) rats. The traces demonstrate substantial cell-to-cell variability of T-current density in both control and STZ-diabetic rats. Electrophysiological recordings were performed at 22–24 °C. The temperature in simulations was set at 23 °C. (**B**) T-currents in a somatic single-compartment DRG neuron model with T-current conductances of 0.14 mS/cm^2^ (black trace, control), 0.20 mS/cm^2^ (red, diabetes), and 0.32 mS/cm^2^ (pink, diabetes). The same voltage clamp stimulation protocol (a depolarization step from −100 to −50 mV) was used in both electrophysiological recordings and simulations. (**C**) APs evoked in the single-compartment model by a short 1 ms threshold stimulation at *G_CaT_* = 0.14, 0.20, and 0.32 mS/cm^2^. A color coding of traces matches one in (**B**). Note that an increase in *G_CaT_* from 0.14 mS/cm^2^ (black trace) to 0.20 and 0.32 mS/cm^2^ results in the generation of one (red trace) and two (pink trace) additional spikes, respectively, as was observed in caps- neurons of STZ-diabetic animals [8]. (**D**) Simulated currents developed during the generation of single spikes (black trace) and double spike bursts (red trace) in STZ-diabetic conditions. Currents during spike generation in control and diabetic conditions are shown in black and red, respectively. 2nd panel—Na-current; 3rd panel—K-current; 4th panel—CaT-current; 5th panel—CaL-current. Note that increased CaT-current in the diabetic conditions (4th panel, green arrow) led to depolarization and generation of a second AP (upper 1st panel, black arrow). (**E**,**F**) Further increase in *G_CaT_* in a single-compartment model resulted in spontaneous AP bursting ((**E**), *G_CaT_* = 1.4 mS/cm^2^) or continuous tonic AP firing ((**F**), *G_CaT_* = 2.2 mS/cm^2^).

**Figure 4 biomolecules-13-01320-f004:**
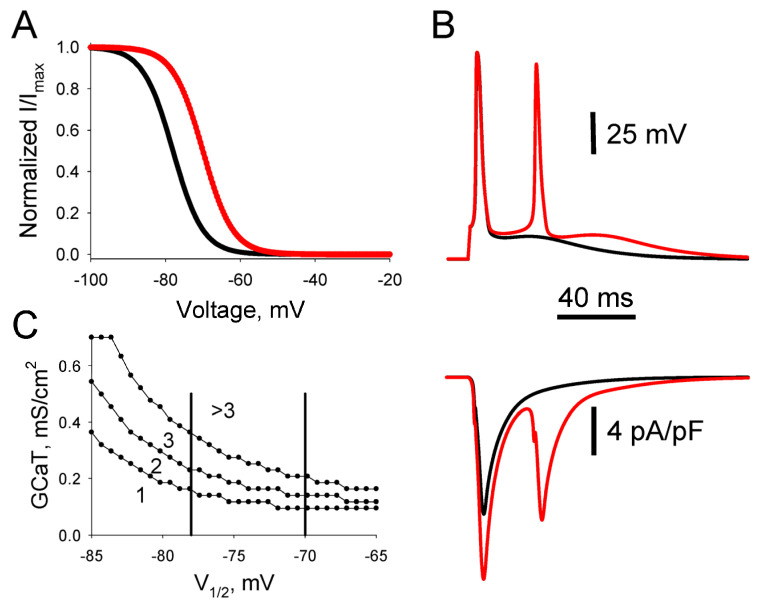
Diabetes-induced shift of T-current inactivation results in increased excitability of nociceptive DRG neurons. (**A**) Steady-state inactivation curves of T-current in control (*V*_1/2_ = −78 mV, black) and diabetic (*V*_1/2_ = −70 mV, red) conditions. (**B**) APs evoked in the single-compartment model by a 1 ms supra-threshold stimulation at *V*_1/2_ = −78 mV (black) and *V*_1/2_ = −70 mV (red); top row. *G_CaT_* = 0.14 mS/cm^2^ for both cases. Simulated CaT-current developed during the generation of a single spike in control (black trace) and double spike burst (red trace) in diabetic conditions (bottom row). Electrophysiological recordings of steady-state inactivation were performed at 22–24 °C [7,21,22]. Temperature in simulations was set at 23 °C. (**C**) A phase diagram showing the number of APs generated at certain values of *V*_1/2_, and conductivity densities of T-current, *G_CaT_*. Each curve represents a boundary at which the number of generated APs is changed by 1. The numbers indicate the quantity of APs generated in the respective phase space. Vertical lines limit the phase space of mean *V*_1/2_ observed in experiments.

**Figure 5 biomolecules-13-01320-f005:**
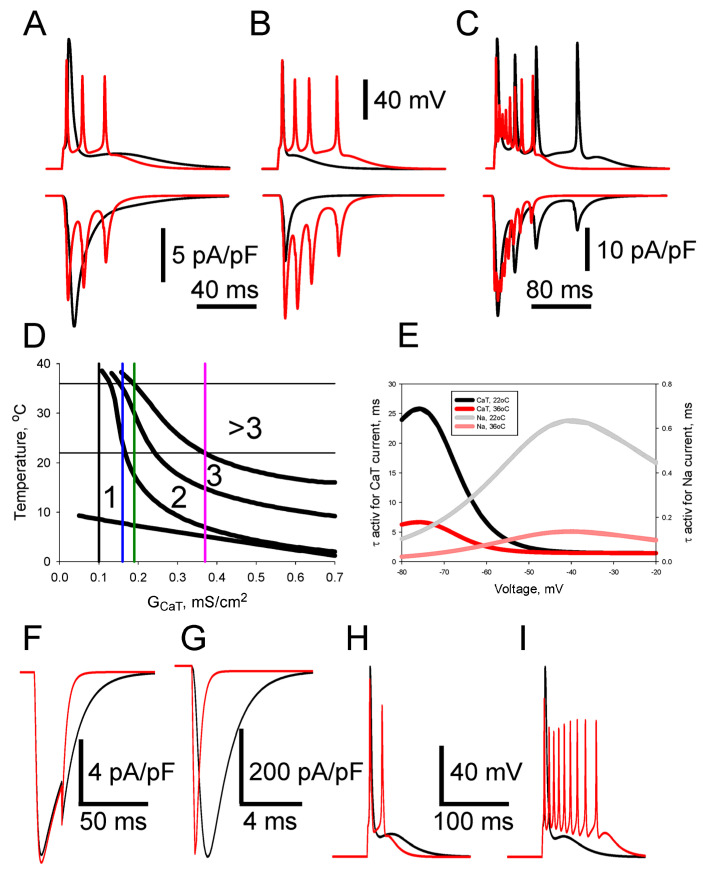
Excitability of nociceptive DRG neurons under diabetic conditions is strongly enhanced at normal body temperature compared to ambient one. (**A**) APs evoked in the single-compartment model by a 1 ms supra-threshold stimulation at 22 °C (black trace) and 36 °C (red trace) (*V*_1/2_ = −78 mV for this and other simulations and *G_CaT_* = 0.16 mS/cm^2^); top row. Simulated CaT-current developed during generation of single spike at 22 °C (black trace) and spike burst at 36 °C (red trace); bottom row. The respective points in a phase diagram (**D**) are located at the intersections of black horizontal lines and vertical blue lines. Note that a conductance of 0.16 mS/cm^2^ represents mild diabetic conditions. (**B**) APs evoked at 36 °C at *G_CaT_* = 0.10 mS/cm^2^ (black trace) and 0.19 mS/cm^2^ (red trace); top row. Simulated CaT-currents developed during generation of single spike (black trace) and spike burst (red trace); bottom row. Note that *G_CaT_* = 0.10 mS/cm^2^ and *G_CaT_* = 0.19 mS/cm^2^ represent the control and diabetic conditions, respectively. The respective points in a phase diagram (**D**) are located at the intersections of the top black horizontal line and vertical black and green lines, respectively. (**C**) APs evoked at *G_CaT_* = 0.37 mS/cm^2^ at 22 °C (black trace) and 36 °C (red trace); top row. Simulated CaT-current developed during generation of spike bursts at 22 °C (black trace) and 36 °C (red trace); bottom row. The respective points in a phase diagram (**D**) are located at the intersections of black horizontal lines and vertical pink lines. Note a substantial difference in patterns of AP bursts. (**D**) A phase diagram showing the number of APs generated at certain values of CaT-current, *G_CaT_*, and temperature. Each curve represents a boundary, at which the number of generated APs is changed by 1. The numbers indicate the quantity of APs generated in the respective phase space. Intersections of vertical and horizontal lines indicate points in the phase space described in (**A**–**C**). (**E**). Voltage-dependence of activation time constants for CaT and Na conductance at 22 °C and 36 °C. Black and red traces represent dependencies for CaT-current at 22 °C and 36 °C, respectively, while grey and pink traces represent ones for Na at 22 °C and 36 °C, respectively. Note that in a range of membrane potentials important for a spike generation, the Na conductance activation time constant is much more sensitive to temperature than one for CaT-current. (**F**). CaT-current simulated in a voltage-clamp mode in response to a step from *V_hold_* = −80 mV to *V_test_* = −20 mV at 22 °C (black trace) and 36 °C (red trace), respectively. Note almost equal activation and amplitudes of the current and its faster tail current at 36 °C. (**G**). Na-current simulated in a voltage-clamp mode in response to a voltage step from *V_hold_* = −80 mV to *V_test_* = −20 mV at 22 °C (black trace) and 36 °C (red trace), respectively. Note the much faster kinetics of Na-current at 36 °C. (**H**). APs evoked in the single-compartment model (with the same time constant for activation of Na-current) by a 1 ms supra-threshold stimulation at 22 °C (black trace) and 36 °C (red trace). The time constant for activation of Na-current in both simulations was set to its value at 22 °C. Compare two APs generated at these conditions at 36 °C with three APs in (**A**) when the time constant for activation of Na-current was set to its value at 36 °C. (**I**) APs evoked with the same time constant for activation of CaT-current at 22 °C (black trace) and 36 °C (red trace). The time constant for activation of CaT-current was set to its value of 22 °C. Compare a large burst of APs generated in these conditions at 36 °C with three APs in (**A**) when the time constant for activation of CaT-current was set to its value at 36 °C. This difference is due to the faster kinetics of the Na-current shown in (**G**) and the longer deactivation of the CaT-current at 22 °C as shown in (**F**). *G_CaT_* was set to 0.16 mS/cm^2^ for (**F**–**I**).

**Figure 6 biomolecules-13-01320-f006:**
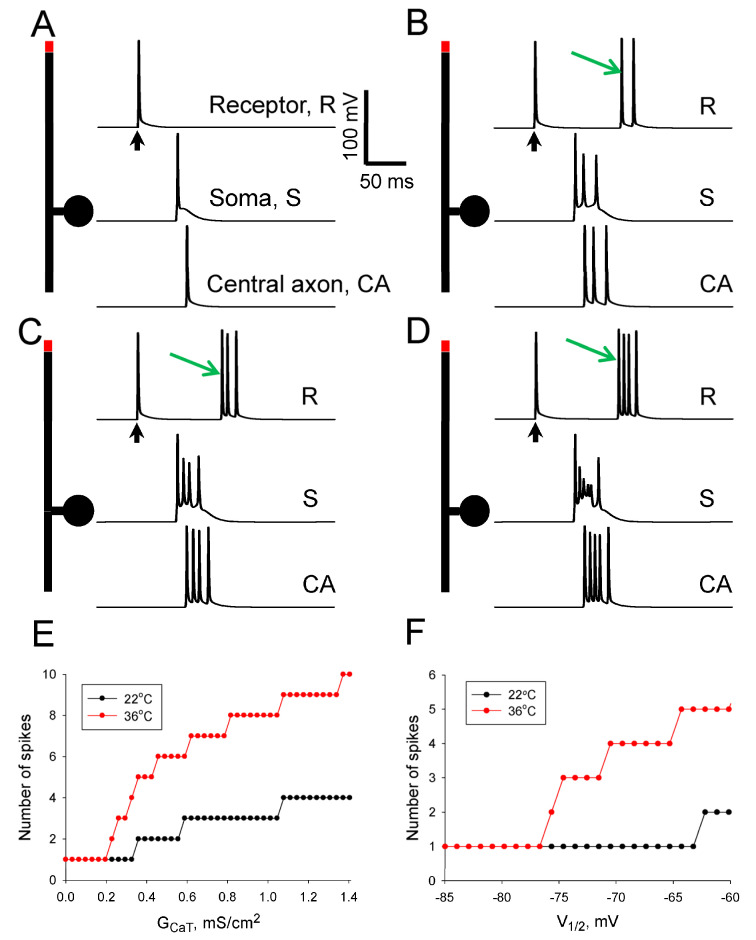
Diabetes-induced somatic amplification of centrally propagated APs evoked in the peripheral axon of nociceptive DRG neurons. (**A**) AP propagation from a receptor zone of the peripheral process to central terminals of nociceptive DRG neurons represented by a multi-compartment model under normal conditions (*G_CaT_* = 0.15 mS/cm^2^; *V*_1/2_ = −78 mV; t = 36 °C). AP evoked in a receptor zone (red rectangle in the upper part of neuron drawing) is shown in the upper trace. This AP propagates along the peripheral axon, invades the soma (middle trace), and further propagates to the end of the central axon (lower trace). A black arrow here and in (**B**–**D**) indicates a moment when a threshold current was applied to the receptor zone. (**B**–**D**). AP propagation from a receptor zone of the peripheral process to central terminals under different diabetic conditions ((**B**): *G_CaT_* = 0.25 mS/cm^2^; *V*_1/2_ = −78 mV; t = 36 °C; (**C**): *G_CaT_* = 0.2 mS/cm^2^; *V*_1/2_ = −70 mV; t = 36 °C; (**D**): *G_CaT_* = 0.25 mS/cm^2^; *V*_1/2_ = −70 mV; t = 36 °C). AP evoked in a receptor zone propagates along the peripheral process, invades the soma, and is amplified by somatic T-channels (middle trace). These APs leave the soma, and propagate both peripherally (indicated by green arrows in the upper traces) and centrally following the first directly propagated APs (lower trace). (**E**) Dependence of the number of APs reaching the end of central axons on conductance density of CaT-current at 22 °C (black) and 36 °C (red) (*V*_1/2_ = −78 mV). Note the steep dependence observed in a range of *G_CaT_* of 0.2–0.4 mS/cm^2^. (**F**) Dependence of the number of spikes reaching the end of central axons on half-maximal steady-state inactivation of CaT-current, *V*_1/2_, at 22 °C (black) and 36 °C (red) (*G_CaT_* = 0.2 mS/cm^2^).

## Data Availability

The data are available upon reasonable request.

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
