# Peer review of "Diabetes-Induced Amplification of Nociceptive DRG Neuron Output by Upregulation of Somatic T-Type Ca^2+^ Channels"

_biomolecules, 2023, doi:10.3390/biom13091320_

Round 1

Reviewer 1 Report

This is an excellent study that used computer modeling to infer the role of Cav3.2 channel alterations in painful peripheral diabetic neuropathy and their impact on cellular excitability. The authors have nicely circumvented limitations of previous studies with isolated DRG cell somas in vitro done at room temperature. The study is well done and nicely presented, it was a pleasure to read.

Please correct typo on page 17, line 610 (first word) 

woful befit from minor editing 

Author Response

This is an excellent study that used computer modeling to infer the role of Cav3.2 channel alterations in painful peripheral diabetic neuropathy and their impact on cellular excitability. The authors have nicely circumvented limitations of previous studies with isolated DRG cell somas in vitro done at room temperature. The study is well done and nicely presented, it was a pleasure to read.

Please correct typo on page 17, line 610 (first word) 

We greatly appreciate the positive evaluation of our MS by the reviewer. We have corrected the typo in the line 610 (now line 617).

Comments on the Quality of English Language: woful befit from minor editing 

We have re-checked the MS and introduced some minor changes to have the same terminology throughout the text of the MS. Some changes have been also introduced based on the suggestions of the other reviewer. These changes are indicated in red in the text of the revised MS.

Reviewer 2 Report

Strength:

In this study, the authors utilized a computational model to elucidate how increased T-type Ca2+ channel expression in DRG contributes to peripheral diabetic neuropathy. Authors clarified that neuronal excitability increased in diabetes-induced T-type Ca2+ channel upregulation at 36 °C than 22°C . In addition, upregulations of somatic T-type Ca2+ channel in diabetes amplified the number of the action potential generated and propagated in a receptor zone of peripheral. These results may contribute to understand the role of T-type Ca2+ channel in peripheral diabetic neuropathy.

Comment:

              The result of this study is clear-cut. I don't have objection in the results of this study.

Minor suggestion:

1. Please check the typographical error, such as MgCl2 at line 102.

2. In addition, it should be showed the reason why the value of T-type Ca2+ channel conductance was varied in each computational model.

3. In diabetic model, is it true that the steady-state inactivation curve of T-type Ca2+ channel shift significantly in Fig.4? It seems that previous study did not show the significant change of V1/2 of steady-state inactivation curve of T-type Ca2+ channel (Duzhyy, D.E et al. Mol Pain. 2015).

It may be better to check the typographical error.

Author Response

In this study, the authors utilized a computational model to elucidate how increased T-type Ca2+ channel expression in DRG contributes to peripheral diabetic neuropathy. Authors clarified that neuronal excitability increased in diabetes-induced T-type Ca2+ channel upregulation at 36 °C than 22°C . In addition, upregulations of somatic T-type Ca2+ channel in diabetes amplified the number of the action potential generated and propagated in a receptor zone of peripheral. These results may contribute to understand the role of T-type Ca2+ channel in peripheral diabetic neuropathy.

Comment:

The result of this study is clear-cut. I don't have objection in the results of this study.

We sincerely thank the reviewer for the positive evaluation of our work. Our response to the reviewer’s comments is given below, Changes/additions to the MS are indicated in red in the text of the revised MS.

Minor suggestion:

  1. Please check the typographical error, such as MgCl2 at line 102.

We have edited the typographical error in line 102.

  1. In addition, it should be showed the reason why the value of T-type Ca2+channel conductance was varied in each computational model.

Thank you for the comment. The values of conductance and V1/2 were varied in the simulations to demonstrate how changes of these parameters experimentally feasible in control and diabetic conditions may affect the excitability of nociceptive prototype neurons. The respective sentence is now added to the text of the MS (lines 436-438).

  1. In diabetic model, is it true that the steady-state inactivation curve of T-type Ca2+channel shift significantly in Fig.4? It seems that previous study did not show the significant change of V1/2of steady-state inactivation curve of T-type Ca2+ channel (Duzhyy, D.E et al. Mol Pain. 2015).

You are right that ‘previous study did not show the significant change of V1/2 of steady-state inactivation curve of T-type Ca2+ channel (Duzhyy, D.E et al. Mol Pain. 2015)’. Simulations related to the shift of the steady-state inactivation (part 3.2.2 of the Results) were conducted for the capsaicin-sensitive IB4 positive nociceptive DRG (caps+) rather than caps- neurons. This shift is clearly shown for this neuronal type (Khomula et al. Specific Functioning of Cav3.2 T-Type Calcium and TRPV1 Channels under Different Types of STZ-Diabetic Neuropathy. Biochim. Biophys. Acta 2013, 1832, 636–649, doi:10.1016/j.bbadis.2013.01.017). It seems that it was not explicitly stated in the part 3.2.2 of the Results although discussed in the part 4.2. In the revised version of the MS, we have edited the part 3.2.2 (lines 408-415) to indicate that the simulations were conducted for the other cell type.

Comments on the Quality of English Language: It may be better to check the typographical error.

We have checked typos in the MS